# Thermodynamic Analysis of Transcritical CO_2_ Ejector Expansion Refrigeration Cycle with Dedicated Mechanical Subcooling

**DOI:** 10.3390/e21090874

**Published:** 2019-09-08

**Authors:** Ruansong Fu, Jinhui Wang, Minfeng Zheng, Kaihong Yu, Xi Liu, Xuelai Li

**Affiliations:** 1College of Chemical Engineering, Fuzhou University, Fuzhou 350116, China (R.F.) (J.W.) (K.Y.); 2College of Ecological Environment and Urban Construction, Fujian University of Technology, Fuzhou 350118, China; 3State Key Laboratory of Photocatalysis on Energy and Environment, Fuzhou University, Fuzhou 350116, China

**Keywords:** transcritical CO_2_ cycle, thermoelectric subcooling, mechanical subcooling, ejector

## Abstract

The new configuration of a transcritical CO_2_ ejector expansion refrigeration cycle combined with a dedicated mechanical subcooling cycle (EMS) is proposed. Three mass ratios of R32/R1234ze(Z) (0.4/0.6, 0.6/0.4, and 0.8/0.2) were selected as the refrigerants of the mechanical subcooling cycle (MS) to further explore the possibility of improving the EMS cycle’s performance. The thermodynamic performances of the new cycle were evaluated using energetic and exergetic methods and compared with those of the transcritical CO_2_ ejector expansion cycle integrated with a thermoelectric subcooling system (ETS). The results showed that the proposed cycle presents significant advantages over the ETS cycle in terms of the ejector performance and the system energetic and exergetic performances. Taking the EMS cycle using R32/R1234ze(Z) (0.6/0.4) as the MS refrigerant as an example, the improvements in the coefficient of performance and system exergy efficiency were able to reach up to 10.27% and 15.56%, respectively, at an environmental temperature of 35 °C and evaporation temperature of −5 °C. Additionally, the advantages of the EMS cycle were more pronounced at higher environmental temperatures.

## 1. Introduction

The critical impact of the refrigeration and air-conditioning industry on ozone depletion and global warming has developed into an alarming issue, which is mainly attributed to the wide application of synthetic refrigerants. Thus, interest in natural refrigerants have been increasing rapidly. In this situation, CO_2_ is regarded to be an extremely promising refrigerant thanks to its zero ODP, negligible GWP, non-toxicity, non-flammability, and superior transport and thermophysical properties [1,2,3]. However, a transcritical cycle mode is generally required at common refrigeration conditions because of the lower critical temperature of CO_2_, which leads to great throttling loss [4]. Consequently, the coefficient of performance (COP) of a transcritical CO_2_ system is often lower than that of a system using synthetic refrigerants. Furthermore, the COP drops rapidly as environmental temperature rises, which further limits the application of the cycle in warm and hot regions [5]. In order to improve system performance, some strategies have been adopted such as employing an expansion device to replace the throttling valve and installing a subcooling system behind the gas cooler.

Using an expansion component instead of a throttling valve is recognized as an extremely effective measure to improve the efficiency of transcritical CO_2_ cycles [6,7,8]. In recent years, both the expander and the ejector have received extensive attention as expansion devices [9,10,11,12,13]. The use of an expander recovers expansion loss and increases the cooling capacity (*Q*_c_), thereby enhancing system performance. However, the efficiency of the CO_2_ expander is low, and the technology is still under maturation [14,15]. The ejector is a promising expansion device due to the fact of its simplicity, reliability, low cost, and high energy-saving effect [16]. Pérez-García et al. [17] conducted a comprehensive comparison of the transcritical CO_2_ cycle with an internal heat exchanger, ejector, and a turbine. The results showed that the system with an ejector exhibited the maximum performance improvement with the increase of evaporation temperature. Deng et al. [18] theoretically studied the transcritical CO_2_ ejector expansion cycle, and an improvement of 22.0% in maximum cooling COP and an increasement of 11.5% in cooling capacity were obtained. An advanced exergy analysis was performed by Bai et al. [19], showing that the exergy destruction was mainly endogenous and 43.44% could be eliminated with the component improvements. Lucas et al. [20] presented an experimental comparison between this system and the conventional throttling valve system and reported that a COP enhancement of 17% was achieved with the assistance of the ejector. Lee et al. [21] experimentally explored the effect of ejector geometry on the CO_2_ air conditioning system performance. The results revealed that there were optimal design parameters in each test and the COP of the cycle using an ejector was increased by 15% relative to the conventional cycle. Li et al. [22] modified the transcritical CO_2_ ejector expansion cycle with a feed-back valve, concluding that a COP improvement of 16% could be achieved under typical air conditioner working conditions.

The use of a subcooling device can decrease the optimum discharge pressure (*p*_opt_) and increase the cooling capacity for the conventional transcritical CO_2_ cycle [23]. In recent years, internal heat exchangers, thermoelectric subcooling, and mechanical subcooling have been relatively effective subcooling methods. The internal heat exchanger is conducive to performance improvement of the conventional throttling valve cycle, while the combination with an expander or an ejector may degrade the system performance [13,24]. Moreover, the application of an internal heat exchanger results in the deterioration of the compressor’s performance and an increase in discharge temperature. Hence, the above disadvantages should be considered in the adoption of an internal heat exchanger. Thermoelectric cooling uses electrical power to produce a lower temperature at the cold side for subcooling the CO_2_ leaving the gas cooler and, thus, there are no such problems. Sarkar [25] performed multi-parameter optimization to get the maximum COP for the transcritical CO_2_ cycle with the thermoelectric subcooling system. The results showed that COP enhancement and discharge pressure drop could reach 25.6% and 15.4%, respectively. An experimental study of this cycle performed by Schoenfield et al. [26] showed that a COP enhancement of 3.3% and a *Q*_c_ improvement of 7.9% over the conventional CO_2_ cycle could be obtained. Dai et al. [27] conducted a performance analysis of transcritical CO_2_ cycles integrated with an expander and a thermoelectric subcooler. The new cycle presented a significant advantage in terms of COP and *p*_opt_, especially at higher gas cooler outlet temperatures or lower evaporation temperatures. Dedicated mechanical subcooling refers to the addition of an auxiliary vapor compression cycle to subcool the fluid from condenser or gas cooler. Llopis et al. [28] theoretically studied the transcritical CO_2_ cycle with dedicated mechanical subcooling, observing that the improvements in COP and *Q*_c_ could reach 20% and 28.8% at the most, respectively. Additionally, the improvement effect was more prominent for environmental temperatures above 25 °C and, thus, this cycle was recommended for warm and hot regions. Llopis et al. [29] experimentally verified the feasibility of this scheme, claiming that this the structure of the CO_2_ cycle was one of the best modifications so far. For better temperature matching, Dai et al. [30] proposed the possibility of using a zeotropic mixture as the MS refrigerant on the basis of the principle of the Lorenz cycle. The effect of different refrigerant mixtures was evaluated and it was found that R32/R1234ze(Z) (0.55/0.45) as an MS refrigerant enhanced COP by 4.91% and decreased *p*_opt_ by 0.11 MPa at *T*_0_ = 35 °C and *T*_e_ = −5 °C. Gullo et al. [31] performed an energetic and environmental analysis on seven CO_2_ booster refrigeration systems and compared them with the cascade cycle. The results illustrated that it is necessary to adopt mechanical subcooling in hot climates. Purohit et al. [32] theoretically evaluated four supermarket refrigeration cycles including a CO_2_ booster cycle with mechanical subcooling. It was found that the configuration with mechanical subcooling exhibited higher energy efficiency than that with parallel compression at higher *T*_0_. Additionally, the annual energy consumption of this subcooling cycle could be reduced by 8.9% in Teheran relative to a R404A multiplex expansion system.

Based on the above literature, it is observed that using an ejector as the expansion device and installing a subcooling device behind the gas cooler are extremely valid ways to enhance the cycle performance. Furthermore, the subcooling device can narrow down the working conditions of the ejector, which simplifies its design or regulation [23]. Liu et al. [33] proposed the combination of a thermoelectric subcooling system and an ejector (ETS) to improve the energy efficiency of the transcritical CO_2_ cycle. In addition, the ETS cycle exhibited prominent advantages in COP_c_ and discharge pressure in the range studied. However, the performance of the thermoelectric subcooler decreases quickly with the increased temperature lift and, thus, a higher subcooling degree results in lower efficiency. The study integrating a mechanical subcooling cycle with a transcritical CO_2_ ejector expansion cycle has not been found, and the effects of thermoelectric subcooling and mechanical subcooling in the CO_2_ cycle have not been compared in detail. Therefore, the transcritical CO_2_ ejector expansion refrigeration cycle combined with the dedicated mechanical subcooling cycle (EMS) is proposed in this study. In addition to pure R32 and R1234ze(Z), three mass ratios of R32/R1234ze(Z) (0.4/0.6, 0.6/0.4, and 0.8/0.2) exhibiting different temperature glides were selected as the MS refrigerants to achieve a good temperature match in the subcooler under different working conditions, thereby further improving the EMS cycle performance. For the purpose of verifying the feasibility of this scheme, energetic and exergetic analysis on the new proposed EMS cycle were carried out, and a comprehensive comparison was also conducted with the ETS cycle.

## 2. Cycle Description and Simulation Model

### 2.1. Cycle Description

Figure 1a depicts the schematic diagram of the ETS cycle, which has been described in detail by Liu et al. [33]. As indicated in Figure 1b, the EMS cycle consists of a transcritical CO_2_ ejector expansion cycle and a mechanical subcooling cycle, where they are connected by a subcooler. The ejector is the main system component including a primary nozzle, a suction chamber, a mixing chamber, and a diffuser. The subcooler is the evaporation component of the mechanical subcooling cycle for subcooling the CO_2_ entering the primary nozzle. For the condenser, the heat transfer fluid is air. Figure 2a,b presents the *p*–*h* and *T*–*s* diagrams of the transcritical CO_2_ ejector expansion cycle with subcooling, respectively. For the EMS cycle, the working principle is illustrated as follows: one-unit mass of saturated vapor (point 1) is compressed into the high-temperature and -pressure vapor (point 2). The vapor entering the gas cooler is isobarically cooled to point 3. The vapor entering the subcooler is further cooled to *T*_3′_ (point 3′). Then the vapor as the primary fluid flows into the primary nozzle and expands into low-pressure and high-velocity fluid (point 4). Simultaneously, *μ* unit mass of secondary flow from the evaporator (point 9) is entrained. The two fluids are mixed at constant pressure, and then the mixed fluid (point 5) flows into the diffuser to obtain a pressure rise. The two-phase fluid from the ejector (point 6) enters the separator where it is divided into saturated liquid and vapor (points 7 and 1). The saturated liquid flows into the evaporator to generate a cooling capacity after the throttling process. The saturated vapor reenters the compressor.

### 2.2. Simulation Model

#### 2.2.1. Assumption of the Model

The model of the EMS cycle is established on the basis of the following assumptions [19,30]:(1)The cycle operates at a steady state and the flow in the ejector is one-dimensional.(2)The heat losses and pressure drops in the pipelines and heat exchangers are negligible.(3)In the CO_2_ cycle, the outgoing flows of the evaporator and separator are saturated. In the mechanical subcooling cycle, the refrigerants leaving the subcooler and condenser are saturated.(4)The primary and secondary flows are mixed at constant pressure. The inlet and outlet velocities in the ejector are neglected.(5)The ejector efficiencies are assumed as constant values. The mechanical efficiency and electrical efficiency of the compressors are constant, and their isentropic efficiency is the expression of pressure ratio.(6)The gas cooler, subcooler, condenser, and evaporator are counter-flow heat exchangers.

The rated operation conditions are presented in Table 1 [30,34]. According to these assumptions and operation conditions, the conservation equations are established, and the cycle performance is evaluated.

#### 2.2.2. Ejector Model

The ejector plays an important role in system performance and, thus, constructing a proper ejector model is crucial for theoretical studies. Generally, the entrainment ratio (*μ*) and pressure lift ratio (*r*_pj_) representing the ejector performance are written as:(1)$$\mu =m_{9}/m_{4}$$
(2)$$r_{\mathrm{pj}}=p_{6}/p_{9}$$

The primary nozzle efficiency is:(3)$$\eta_{n}=\left(h_{3^{\prime }}-h_{4}\right)/\left(h_{3^{\prime }}-h_{4s}\right)$$

From the viewpoint of energy conservation, the velocity of the primary fluid entering the mixing chamber can be derived as:(4)$$u_{4}=\sqrt{2\left(h_{3^{\prime }}-h_{4}\right)}$$

The secondary fluid inlet velocity *u*_9_ is negligible compared with the primary fluid velocity at the nozzle outlet [35] and, thus, the ideal velocity at the mixing chamber outlet is:(5)$$u_{5}^{\prime }=u_{4}/\left(1+\mu \right)$$

In this study, the friction loss of the mixing process was considered, and the mixing efficiency *η*_mix_ [36] was introduced to obtain the velocity at the mixing chamber outlet.

(6)$$\eta_{\mathrm{mix}}=u_{5}^{2}/{u_{5}^{\prime }}^{2}$$

(7)$$u_{5}=\frac{u_{4}}{1+\mu }\sqrt{\eta_{\mathrm{mix}}}$$

The energy conservation equation in the mixing section is:(8)$$\left(1+\mu \right)\left(h_{5}+\frac{1}{2}u_{5}^{2}\right)=\mu h_{9}+h_{4}+\frac{1}{2}u_{4}^{2}$$

Based on the energy conservation, the specific enthalpy at the diffuser exit is:(9)$$h_{6}=h_{5}+\frac{1}{2}u_{5}^{2}$$
where *h*_5_ and *h*_6_ are the specific enthalpy of the mixed fluid at the diffuser inlet and outlet, respectively.

The isentropic efficiency of the diffuser is:(10)$$\eta_{d}=\left(h_{6s}-h_{5}\right)/\left(h_{6}-h_{5}\right)$$
where *h*_6s_ is the ideal specific enthalpy at the diffuser outlet through an isentropic compression process.

The fluid leaving the ejector is separated into saturated liquid and saturated vapor and the mass flow ratio of the two fluids remains constant in steady-state operation. The entrainment ratio is equal to this ratio and the mass percent of saturated vapor is equal to the vapor quality at the ejector outlet. Therefore, the following equation must be satisfied in a stable system:(11)$$x_{6}=1/\left(1+\mu \right)$$

Compared with *μ* and *r*_pj_, the ejector exergy efficiency can reflect performance more comprehensively, which is defined as:(12)$$\eta_{\mathrm{ej}}=\left[m_{\mathrm{eva}}\left({\mathrm{ex}}_{6}{-\mathrm{ex}}_{9}\right)\right]/\left[m_{\mathrm{sc}}\left({\mathrm{ex}}_{3^{\prime }}{-\mathrm{ex}}_{6}\right)\right]$$

#### 2.2.3. Energetic Analysis

The modeling analysis was conducted on the basis of 1 kg/s CO_2_ flowing into the compressor (point 1). Hence, the mass flow rates of the components in the CO_2_ cycle are expressed as:(13)$$m_{\mathrm{com}}{=m}_{\mathrm{gc}}=m_{\mathrm{sc}}=1$$
(14)$$m_{\mathrm{val}}{=m}_{\mathrm{eva}}=\mu$$
(15)$$m_{\mathrm{ej}}{=m}_{\mathrm{sp}}=1+\mu$$

Based on energy conservation, the mass flow rate of refrigerant in the MS cycle is:(16)$$m_{\mathrm{MS}}{=m}_{\mathrm{sc}}\left(h_{3}-h_{3^{\prime }}\right)/\left(h_{a^{\prime }}-h_{d^{\prime }}\right)$$

The compressor work, cooling capacity, and COP of the MS cycle are expressed, respectively, as:(17)$$W_{\mathrm{MS}}=m_{\mathrm{MS}}\left(h_{b^{\prime }s}-h_{a^{\prime }}\right)/\left(\eta_{\mathrm{me}}\eta_{\mathrm{el}}\eta_{s,\mathrm{MS}}\right)$$
(18)$$Q_{\mathrm{MS}}=m_{\mathrm{MS}}\left(h_{a^{\prime }}-h_{d^{\prime }}\right)$$
(19)$${\mathrm{COP}}_{\mathrm{MS}}=Q_{\mathrm{MS}}/W_{\mathrm{MS}}$$
where *η*_s,MS_ refers to the compressor isentropic efficiency in the MS cycle, expressed as [37]:(20)$$\eta_{s,\mathrm{MS}}=0.83955-0.01026\left(p_{b^{\prime }}/p_{a^{\prime }}\right)-0.00097{\left(p_{b^{\prime }}/p_{a^{\prime }}\right)}^{2}$$

The power consumption of the CO_2_ compressor is:(21)$$W_{{\mathrm{CO}}_{2}}{=m}_{\mathrm{com}}\left(h_{2s}{-h}_{1}\right)/\left(\eta_{\mathrm{me}}\eta_{\mathrm{el}}\eta_{{s,\mathrm{CO}}_{2}}\right)$$
where *η*_s,CO2_ refers to isentropic efficiency of the CO_2_ compressor, written as [38]:(22)$$\eta_{{s,\mathrm{CO}}_{2}}=0.815+0.022\left(p_{2}/p_{1}\right)-0.0041{\left(p_{2}/p_{1}\right)}^{2}+0.0001{\left(p_{2}/p_{1}\right)}^{3}$$

The compressor work, cooling capacity, and COP of the EMS cycle are expressed, respectively, as,
(23)$$W_{\mathrm{tot}}{=W}_{\mathrm{MS}}{+W}_{{\mathrm{CO}}_{2}}$$
(24)$$Q_{c}{=m}_{\mathrm{eva}}\left(h_{9}{-h}_{8}\right)$$
(25)$${\mathrm{COP}}_{c}{=Q}_{c}/W_{\mathrm{tot}}$$

#### 2.2.4. Exergetic Analysis

Exergy analysis has become an extremely effective way to verify system feasibility and maximize system energy savings [39]. The exergy destruction rate of each component is written as follows:

For the compressor in the CO_2_ cycle:(26)$$I_{{\mathrm{com},\mathrm{CO}}_{2}}{=m}_{\mathrm{com}}T_{0}\left(s_{2}{-s}_{1}\right)$$

For the gas cooler:(27)$$I_{{\mathrm{gc},\mathrm{CO}}_{2}}{=m}_{\mathrm{gc}}\left[\left(h_{2}{-h}_{3}\right){-T}_{0}\left(s_{2}{-s}_{3}\right)\right]$$

For the ejector:(28)$$I_{{\mathrm{ej},\mathrm{CO}}_{2}}{=m}_{\mathrm{sc}}T_{0}\left(s_{6}-s_{3^{\prime }}\right){+m}_{\mathrm{eva}}T_{0}\left(s_{6}{-s}_{9}\right)$$

For the throttling valve in the CO_2_ cycle:(29)$$I_{{\mathrm{val},\mathrm{CO}}_{2}}{=m}_{\mathrm{val}}T_{0}\left(s_{8}{-s}_{7}\right)$$

For the evaporator:(30)$$I_{{\mathrm{eva},\mathrm{CO}}_{2}}{=m}_{\mathrm{eva}}T_{0}\left[\left(h_{8}{-h}_{9}\right)/T_{r}-\left(s_{8}{-s}_{9}\right)\right]$$

For the subcooler:(31)$$I_{\mathrm{sc}}{=m}_{\mathrm{sc}}T_{0}\left(s_{3^{\prime }}-s_{3}\right){-m}_{\mathrm{MS}}T_{0}\left(s_{d^{\prime }}-s_{a^{\prime }}\right)$$

For the compressor in the MS cycle:(32)$$I_{\mathrm{com},\mathrm{MS}}{=m}_{\mathrm{MS}}T_{0}\left(s_{b^{\prime }}-s_{a^{\prime }}\right)$$

For the condenser:(33)$$I_{\mathrm{con},\mathrm{MS}}{=m}_{\mathrm{air}}T_{0}\left(s_{\mathrm{air},\mathrm{out}}{-s}_{\mathrm{air},\mathrm{in}}\right){-m}_{\mathrm{MS}}T_{0}\left(s_{b^{\prime }}-s_{c^{\prime }}\right)$$

For the throttling valve in the MS cycle:(34)$$I_{\mathrm{val},\mathrm{MS}\;}{=m}_{\mathrm{MS}}T_{0}\left(s_{d^{\prime }}-s_{c^{\prime }}\right)$$

The total exergy destruction rate of the MS cycle is:(35)$$I_{\mathrm{tot},\mathrm{MS}}{=I}_{\mathrm{sc}}{+I}_{\mathrm{com},\mathrm{MS}}{+I}_{\mathrm{con},\mathrm{MS}}{+I}_{\mathrm{val},\mathrm{MS}}$$

The total exergy destruction rate of the EMS cycle is:(36)$$I_{\mathrm{tot}}{=I}_{{\mathrm{com},\mathrm{CO}}_{2}}{+I}_{{\mathrm{gc},\mathrm{CO}}_{2}}{+I}_{{\mathrm{ej},\mathrm{CO}}_{2}}{+I}_{{\mathrm{val},\mathrm{CO}}_{2}}{+I}_{{\mathrm{eva},\mathrm{CO}}_{2}}{+I}_{\mathrm{sc}\;}{+I}_{\mathrm{com},\mathrm{MS}}{\;+\;I}_{\mathrm{con},\mathrm{MS}\;}{+\;I}_{\mathrm{val},\mathrm{MS}}$$

The exergy efficiency of the EMS cycle is:(37)$$\eta_{\mathrm{ex}}{=\;1-I}_{\mathrm{tot}}/W_{\mathrm{tot}}$$

#### 2.2.5. Solving Procedure

According to the above model, the simulation code was written in MATLAB to study system performance at various working conditions, where the properties of refrigerants and air were obtained using REFPROP 9.1 [40]. Figure 3 presents the flowchart of computational procedure, which is explained in detail as follows:
(1)The properties at points 9, 3, and 3′ can be obtained based on environmental temperature, evaporation temperature, discharge pressure and subcooling degree. The properties at point 4 are calculated through nozzle efficiency and evaporation pressure.(2)The entrainment ratio is assumed.(3)The properties at point 5 are calculated according to Equations (5)–(8) and evaporation pressure. Based on the energy conservation and given diffuser efficiency, the properties at point 6 are obtained.(4)If the condition $\mathrm{Abs}\left(x_{6}-1/\left(1\;+\;\mu \right)\right){<4e}^{-7}$ is not met, *μ* is re-assumed and step (3) is executed again.(5)The properties of points 1, 7, 8, and 2 are calculated.(6)*T*_a′_ and *T*_c′_ are assumed.(7)The properties of points a′, c′, d′, and b′ are calculated according to the mass fraction of R32 and the assumed *T*_a′_ and *T*_c′_.(8)If the absolute deviation between the calculated pinch temperature difference of the condenser and the specified value is greater than 0.001 °C, *T*_c′_ is re-assumed and step (7) is performed.(9)If the condition $\mathrm{Abs}\left({\Delta T}_{\mathrm{sc},\mathrm{pinch},\mathrm{cal}}{-\Delta T}_{\mathrm{sc},\mathrm{pinch}}\right)<0.001\;{}^{\circ}C$ is not satisfied, steps (6)–(8) will be repeated until this condition met.(10)Performance parameters for evaluating the EMS cycle are calculated using the above equations.

## 3. Simulation Results and Discussion

### 3.1. The Optimum Operating Condition

Figure 4 shows COP_c_ variations of the ETS and the EMS cycles with subcooling degree and discharge pressure at *T*_0_ = 35 °C and *T*_e_ = −5 °C when R32/R1234ze(Z) (0.6/0.4) was used as the MS refrigerant. COP_c_ first increased rapidly and then decreased gradually with the discharge pressure, and the effect of the subcooling degree on COP_c_ also showed a similar pattern, which is consistent with the results in previous studies and which have been fully explained [27,41]. Furthermore, a maximize COP_c_ can be obtained with a simultaneous optimization of subcooling degree and discharge pressure. The corresponding optimum subcooling degree and discharge pressure constitute the optimum operating condition. For the purpose of more reasonably comparing the systems, the following analyses were carried out based on the optimum operating condition.

Table 2 exhibits the performance comparison of the ETS and the EMS cycles at optimum subcooling degree and discharge pressure under the given working condition. Although the advantage of the EMS cycle in *p*_opt_ was inconspicuous, the increase in the cooling capacity could reach 23.19% relative to the ETS cycle. More importantly, the improvement in the maximum COP_c_ reached 10.27% and the enhancement in the system exergy efficiency was up to 15.56% compared with the ETS cycle. In addition, remarkable improvements in the subcooling system and the ejector were also achieved, with COP_MS_ and *η*_ej_ increasing by 13.47% and 21.82%, respectively. Therefore, the use of mechanical subcooling in the transcritical CO_2_ ejector expansion refrigeration cycle presented higher performance than thermoelectric subcooling under the optimum operating condition.

### 3.2. Effect of Environmental Temperature

The variations of optimum subcooling degree and discharge pressure with environmental temperature for the ETS and the EMS cycles at *T*_e_ = –5 °C are described in Figure 5. The EMS cycles using mixed refrigerants showed a higher optimum subcooling degree than those using pure refrigerants. The EMS (0.4/0.6) cycle had a larger temperature glide in the subcooler and the corresponding higher optimum subcooling degree, followed by the EMS (0.6/0.4) cycle. The optimum subcooling degree of the EMS (0.8/0.2) cycle exhibited a similar trend to that of the EMS cycle using pure R1234ze(Z) or R32 due to the small temperature glide of R32/R1234ze(Z) (0.8/0.2). The thermoelectric subcooler showed high performance at low-temperature lift and, thus, its optimum subcooling degree was minimal in the range studied. It can be seen from Figure 5b that the optimum discharge pressure increased with the environmental temperature. The optimum discharge pressure of the EMS cycles at both 0.8/0.2 and 0.6/0.4 were lower than that of the ETS cycle, especially for the higher environmental temperatures.

Figure 6 presents the effect of environmental temperature on ejector performance of the ETS and the EMS cycles at *T*_e_ = –5 °C. Both the nozzle inlet temperature and optimum discharge pressure increased with the environmental temperature, which indicates that the vapor quality at the nozzle and diffuser outlets increased, thereby leading to a smaller *μ*. The pressure lift ratio increased with the environmental temperature, which was attributed to the decreased entrainment ratio and the increased nozzle pressure drop. The vapor quality at the nozzle outlet decreased with increasing subcooling degree and, thus, the EMS cycles showed a higher *μ* and a lower *r*_pj_. The ejector exergy efficiency decreased with the increasing environmental temperature, which is mainly explained by the following aspects. The temperature difference between the primary and secondary flows increased with the increasing *T*_0_ and, thus, a higher irreversible loss was obtained in the mixing chamber. The decreased entrainment ratio also reduced ejector exergy efficiency. Increasing the subcooling degree can alleviate the deterioration of the ejector exergy efficiency, so the EMS cycles exhibited higher ejector performance, especially the EMS (0.4/0.6) cycle.

Figure 7 presents the variations of total power consumption, cooling capacity, and maximum COP_c_ with environmental temperature for the ETS and the EMS cycles at *T*_e_ = –5 °C. The increased optimum discharge pressure with higher *T*_0_ resulted in a greater *W*_CO2_. Except for the EMS (0.4/0.6) and the EMS (0.6/0.4) cycles, the optimum subcooling degree of the other four cycles increased with environmental temperature, resulting in a rapid augment in the power consumption of the subcooling system. Therefore, *W*_tot_ of these four cycles presented a large upward trend as shown in Figure 7a. Additionally, the EMS cycles consumed more power than the ETS cycle due to a lower pressure lift ratio and a higher subcooling degree. With the rise of environmental temperature, the increasing pressure lift ratio resulted in an augment in the evaporator inlet enthalpy, thereby reducing the unit cooling capacity. Simultaneously, a lower entrainment ratio was obtained at higher *T*_0_ as shown in Figure 6a. Therefore, the downward trend in cooling capacity was presented with increasing *T*_0_. Compared with the ETS cycle, the EMS cycles achieved a significant enhancement in cooling capacity at the expense of more power consumption. Furthermore, the increment in the cooling capacity was greater than that in the power consumption. Therefore, the EMS cycles presented a higher COP_c_ as shown in Figure 7c. It should be mentioned that the COP_c_ improvement was more pronounced at higher environmental temperatures. For example, the maximum COP_c_ of the EMS (0.6/0.4) cycle was up to 11.77% higher than that of the ETS cycle at *T*_0_ = 40 °C.

It can also be found from Figure 7 that the mass ratio of the MS refrigerant has a large influence on system energetic performance. For purpose of illustrating the effect of the mass ratio on the maximum COP_c_, the variation of exergy destruction rate for each component of the MS cycle using mixed refrigerants with environmental temperature is described in Figure 8. The condenser of the EMS (0.8/0.2) cycle had a lower exergy destruction rate compared with the EMS (0.4/0.6) and the EMS (0.6/0.4) cycles, which is due to that the temperature glide of R32/R1234ze(Z) (0.8/0.2) is relatively close to the air side inlet and outlet temperature difference, and thus a better temperature match is obtained. R32/R1234ze(Z) (0.8/0.2) with lower temperature glide is suitable for providing a small subcooling degree required for the CO_2_ cycle at lower *T*_0_, thereby forming a superior temperature match in the subcooler. Simultaneously, a lower subcooling degree and temperature glide can reduce the temperature difference between the high and low pressure side, resulting in the less irreversibility loss in the throttling valve and compressor. Therefore, the EMS (0.8/0.2) cycle exhibits a higher maximum COP_c_ at lower environmental temperatures. However, the subcooling degree required for the CO_2_ cycle increases with environmental temperature, leading to a sharp deterioration of the MS cycle using R32/R1234ze(Z) (0.8/0.2) as refrigerant. At this point, R32/R1234ze(Z) (0.6/0.4) with higher temperature glide is recommended as the MS refrigerant to achieve a higher COP_c_. The COP_c_ improvement of the EMS (0.4/0.6) cycle relative to the ETS cycle increases rapidly with the rise of environmental temperature, which is due to the fact that R32/R1234ze(Z) (0.4/0.6) with the highest temperature glide is more suitable for high temperature regions. In summary, the mass ratio of the MS refrigerant should be selected based on different working conditions.

The variations of total exergy destruction rate and system exergy efficiency with environmental temperature for the ETS and the EMS cycles at *T*_e_ = −5 °C are represented in Figure 9. The total exergy destruction rate of the ETS cycle approximately linearly increased with environmental temperature, while the growth rate of the EMS cycles was relatively moderate. The total exergy destruction rate of the EMS (0.8/0.2) cycle was always lower than the ETS cycle in the *T*_0_ range studied, and this improvement increased with the environmental temperature. The exergy efficiency of the ETS cycle decreased rapidly with increasing *T*_0_, while the EMS (0.4/0.6) and the EMS (0.6/0.4) cycles presented an upward trend. The exergy efficiency of the EMS cycles had a significant enhancement compared with the ETS cycle, especially at higher *T*_0_. For example, the exergy efficiency of the EMS (0.6/0.4) cycle can reach 17.38% higher than that of the ETS cycle at *T*_0_ = 40 °C. From the perspective of exergetic performance, it is further illustrated that the advantages of the EMS cycles were more pronounced for the higher environmental temperatures. Additionally, the use of mixed refrigerant of a proper mass ratio in the mechanical subcooling cycle presented superior system exergetic performance compared to pure refrigerants.

### 3.3. Effect of Evaporation Temperature

Figure 10 presents the variations of optimum subcooling degree and discharge pressure with evaporation temperature for the ETS and the EMS cycles at an environmental temperature of 35 °C. In the evaporation temperature range studied, the EMS cycles still exhibited a higher optimum subcooling degree relative to the ETS cycle. For the EMS (0.8/0.2), the EMS (0/1), and the EMS (1/0) cycles, the decrease in subcooling degree increased the evaporation temperature (*T*_a’_) of the mechanical subcooling cycle, thereby increasing COP_MS_. Therefore, the optimum subcooling degree of these three cycles decreased with evaporation temperature. Due to the high-temperature glide of R32/R1234ze(Z) (0.4/0.6), the reduction in subcooling degree did not increase *T*_a’_ and improve the performance of the subcooling cycle, but rather resulted in a decrease in the cooling capacity of the CO_2_ cycle. Therefore, the optimum subcooling degree of the EMS (0.4/0.6) cycle did not vary with the evaporation temperature. The trend of the optimum subcooling degree of the EMS (0.6/0.4) cycle can also be well explained by the above reasons. The optimum discharge pressure for the EMS (0.4/0.6) and the EMS (0.6/0.4) cycles increased with the evaporation temperature. However, the evaporation temperature had little effect on the optimum discharge pressure of the other four cycles. The optimum discharge pressure of the EMS (0.8/0.2) cycle was reduced by 0.185–0.235 MPa compared with the ETS cycle at the given working conditions.

Figure 11 shows ejector performance of the ETS and the EMS cycles versus evaporation temperature at *T*_0_ = 35 °C. As the evaporation temperature increased, the vapor quality at the primary nozzle and diffuser outlets decreased, leading to a higher *μ*. A lower pressure lift ratio was achieved at higher evaporation temperatures due to the fact that the primary fluid with a lower nozzle pressure drop entrains more secondary fluid. Compared with the ETS cycle, the EMS cycles exhibited a higher entrainment ratio and a correspondingly lower pressure lift ratio, which has been explained in the description of Figure 6. As shown in Figure 11c, using mechanical subcooling in the transcritical CO_2_ ejector expansion refrigeration cycle presents higher ejector exergy efficiency than thermoelectric subcooling. For example, the ejector exergy efficiency of the EMS (0.4/0.6) cycle was up to 32.57% higher than that of the ETS cycle at *T*_e_ = −5 °C. In addition, the mechanical subcooling cycle working with mixed refrigerant also exhibited higher ejector performance.

The variations of total power consumption, cooling capacity, and maximum COP_c_ of the ETS and the EMS cycles with evaporation temperature at environmental temperature of 35 °C are described in Figure 12. The lower power consumption of the CO_2_ compressor for the ETS cycle was mainly attributed to the larger pressure lift ratio. Simultaneously, its subcooling system also consumed less power due to the lower optimum subcooling degree. Therefore, the total power consumption of the ETS cycle was the smallest in the *T*_e_ range studied. Although the cooling capacity can be improved due to the increased *μ* and the reduced *r*_pj_, the unit cooling capacity decreased rapidly as the evaporation temperature increased. Consequently, the cooling capacity showed a downward trend, as seen in Figure 12b. Compared with the ETS cycle, the EMS cycles had a higher entrainment ratio and a lower pressure lift ratio resulting in a larger cooling capacity. As depicted in Figure 12c, the EMS cycles presented an evident COP_c_ improvement relative to the ETS cycle. When *T*_e_ was lower than −10 °C, the COP_c_’s improvement of the EMS cycles using mixed refrigerants was similar. However, this enhancement of the EMS (0.4/0.6) cycle gradually decreased with increasing *T*_e_, even lower than the EMS cycle using pure refrigerant, and the EMS (0.8/0.2) cycle exhibited an opposite trend. Therefore, it is necessary to select a proper mass ratio of the MS refrigerant according to different working conditions.

Figure 13 shows the variations in the total exergy destruction rate and system exergy efficiency with evaporation temperature for the ETS and the EMS cycles at *T*_0_ = 35 °C. It can be found that the total exergy destruction rate and system exergy efficiency decreased with the increasing evaporation temperature for the ETS and the EMS cycles, and the EMS (0.8/0.2) cycle showed a lower *I*_tot_ and a higher *η*_ex_ than the ETS cycle. When *T*_e_ varied from −30 to 10 °C and *T*_0_ was fixed at 35 °C, the EMS (0.8/0.2) cycle exhibited a 7.20–8.58% lower exergy destruction rate and a 12.87–17.44% higher exergy efficiency over the ETS cycle. In addition, the mass ratio of the mixed refrigerant in the mechanical subcooling cycle also had an effect on the system exergetic performance, especially the total exergy destruction rate.

## 4. Conclusions

The transcritical CO_2_ ejector expansion refrigeration cycle combined with the mechanical subcooling cycle was proposed in this study. Energetic and Exergetic analyses were performed to study the thermodynamic performance of the proposed cycle, and a comprehensive comparison with the ETS cycle was conducted. The major conclusions are summarized as follows:(1)A maximum COP_c_ was achieved for the ETS and the EMS cycles with a simultaneous optimization of subcooling degree and discharge pressure. The EMS cycle showed a higher optimum subcooling degree, and a lower optimum discharge pressure was obtained when the mixed refrigerant with a proper mass ratio was selected in the MS cycle.(2)The new proposed EMS cycle performed well in terms of the ejector performance, the system energetic and exergetic performance. Taking the EMS (0.6/0.4) cycle as an example, the maximum COP_c_ was enhanced by 10.27% and the improvement in the system exergy efficiency was up to 15.56% compared with the ETS cycle at *T*_0_ = 35 °C and *T*_e_ = −5 °C. Additionally, a 23.19% increment in *Q*_c_ and a 21.82% enhancement in *η*_ej_ were also obtained in this condition.(3)It is recommended to apply the new cycle in warm and hot regions, where the improvements in COP_c_ and *η*_ex_ were more obvious.(4)It is necessary to select a proper mass ratio of the mixed refrigerant in the mechanical subcooling cycle according to different working conditions.

## Figures and Tables

**Figure 1 entropy-21-00874-f001:**
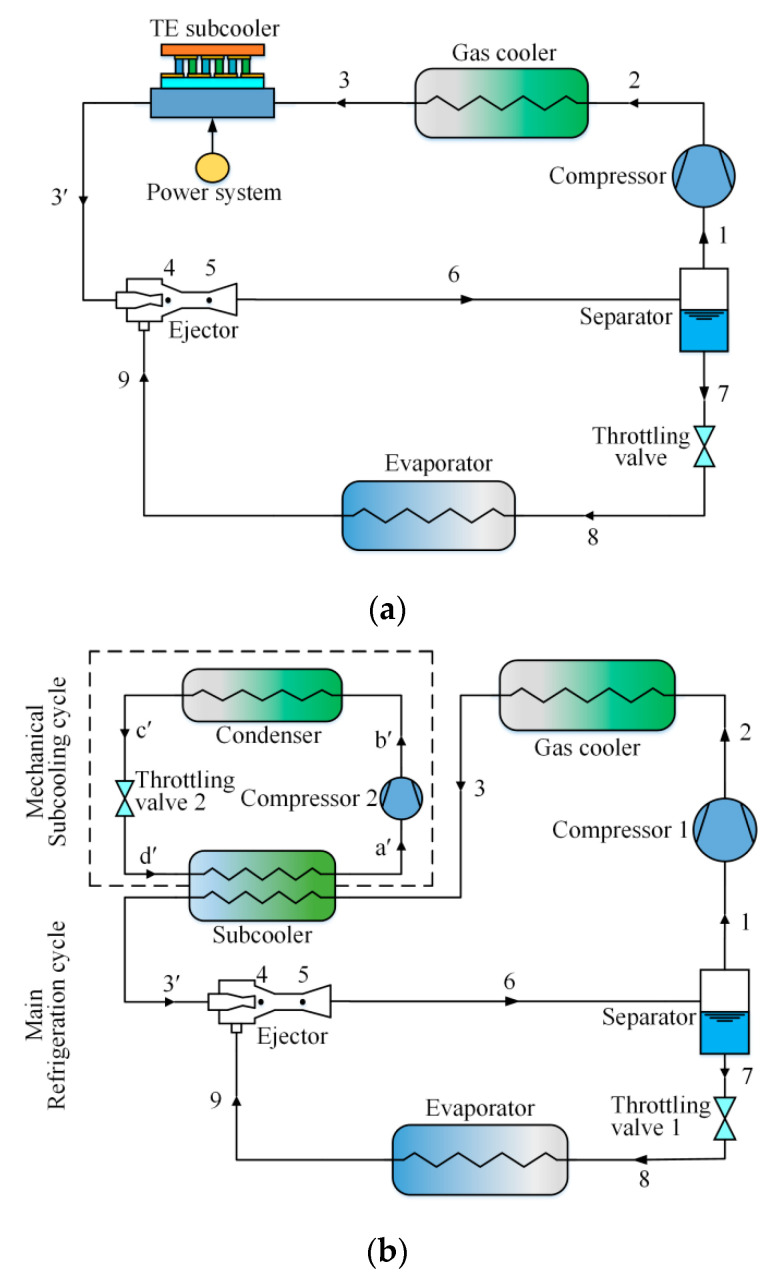
Schematic diagrams: (**a**) The ETS cycle; (**b**) The EMS cycle.

**Figure 2 entropy-21-00874-f002:**
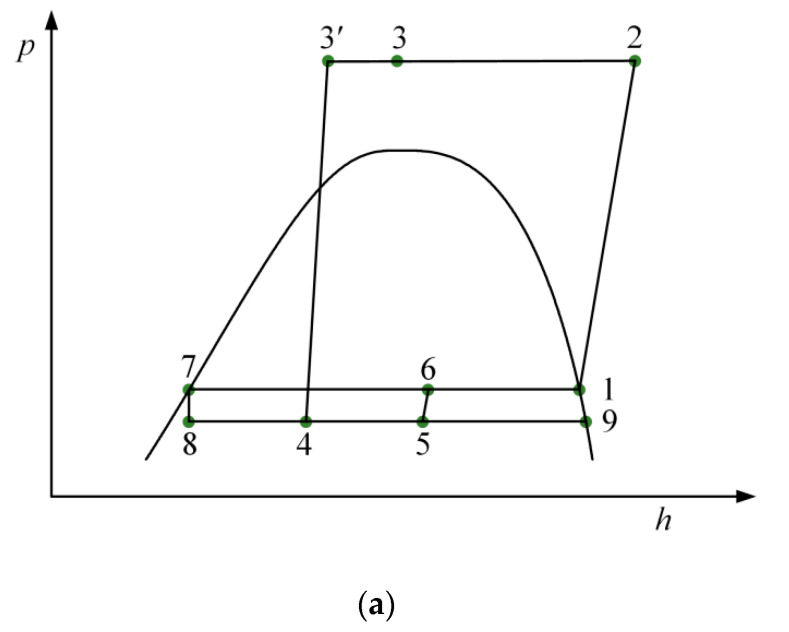
The transcritical CO_2_ ejector expansion cycle with subcooling: (**a**) *p*–*h* diagram; (**b**) *T*–*s* diagram.

**Figure 3 entropy-21-00874-f003:**
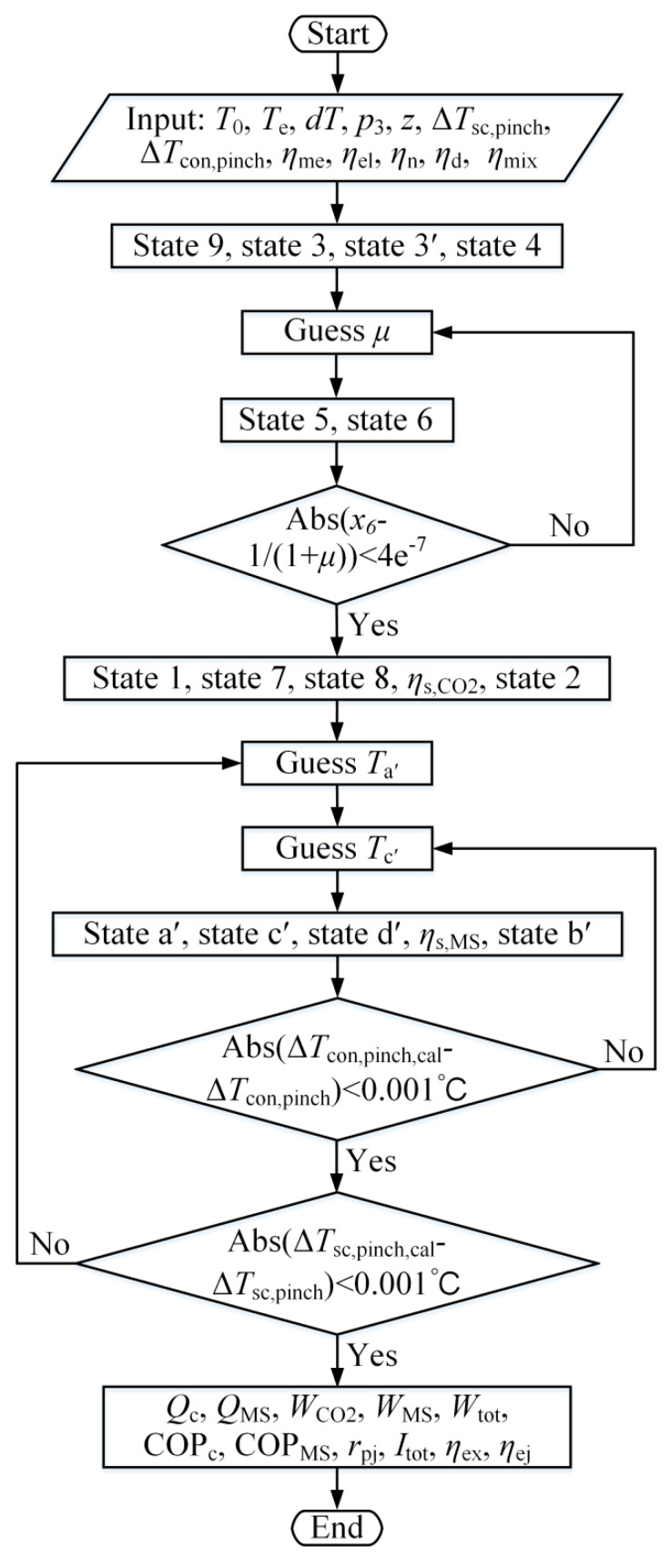
Flowchart of computational procedure.

**Figure 4 entropy-21-00874-f004:**
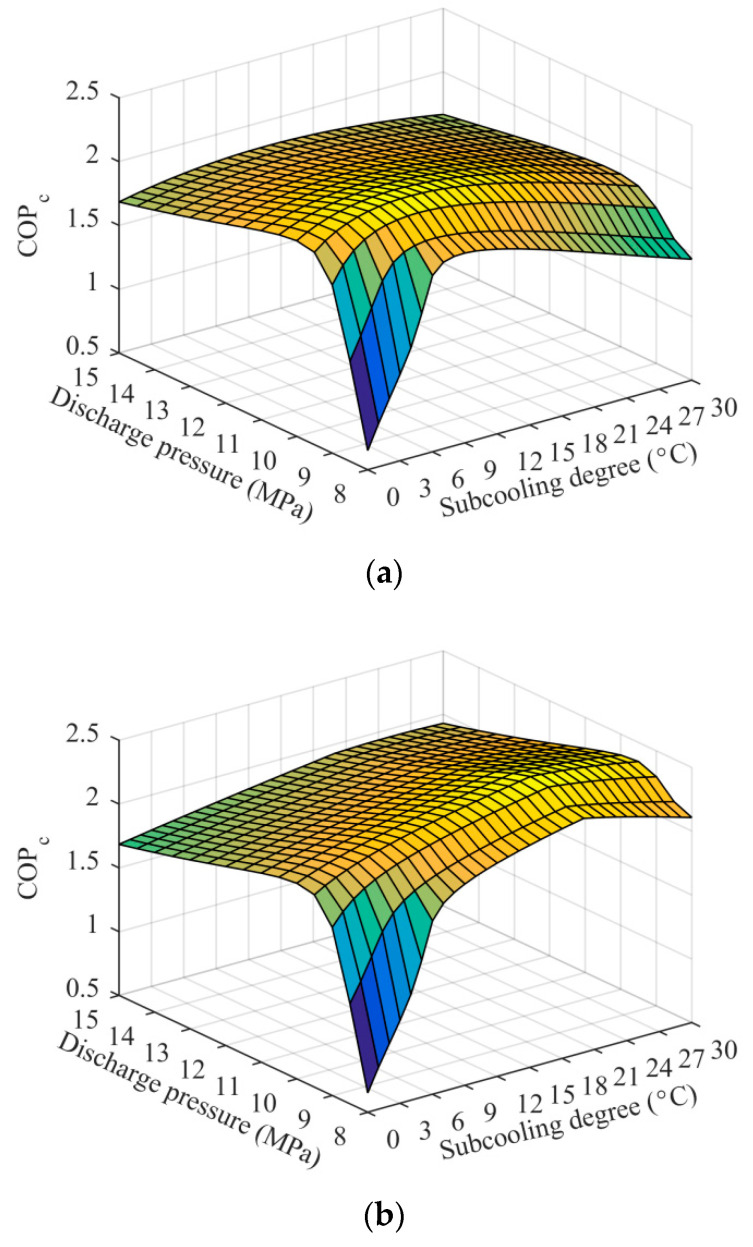
The variation of COP_c_ with subcooling degree and discharge pressure: (**a**) ETS cycle; (**b**) EMS cycle.

**Figure 5 entropy-21-00874-f005:**
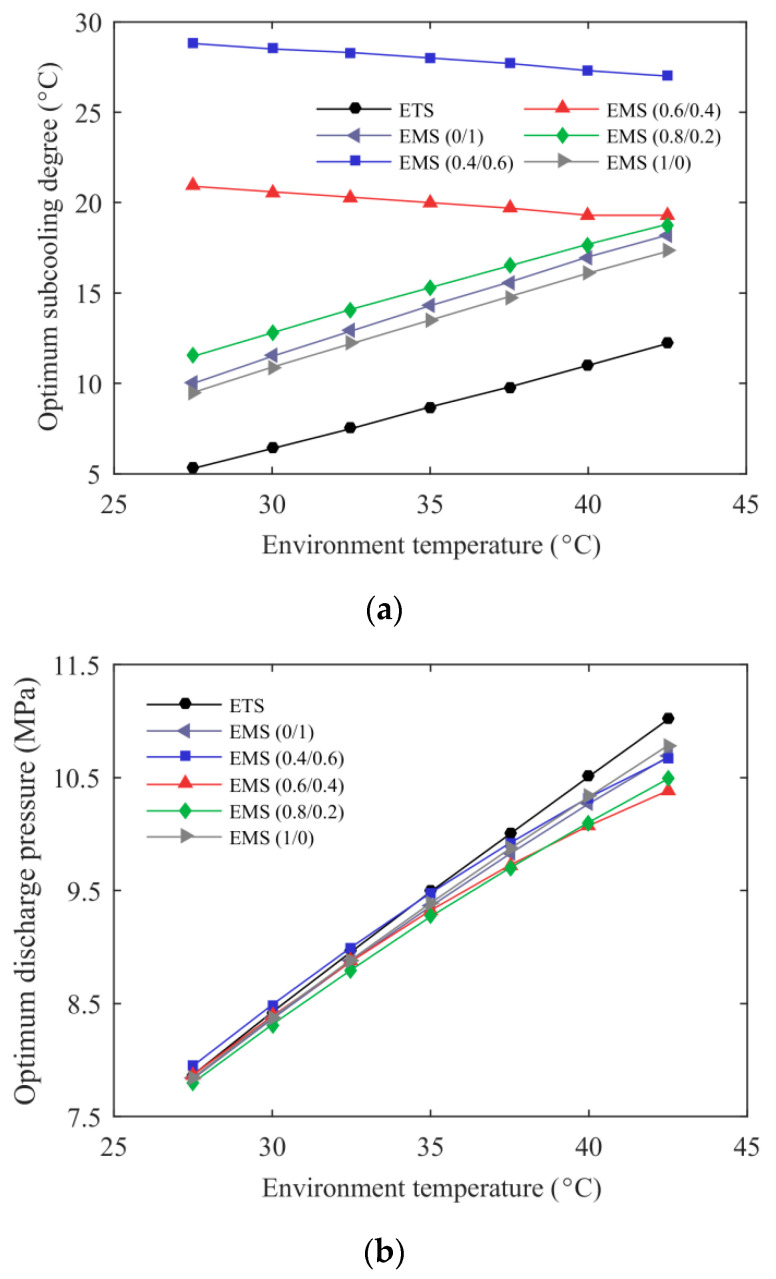
Optimum operating condition of the ETS and the EMS cycles’ variation with environmental temperature: (**a**) optimum subcooling degree; (**b**) optimum discharge pressure.

**Figure 6 entropy-21-00874-f006:**
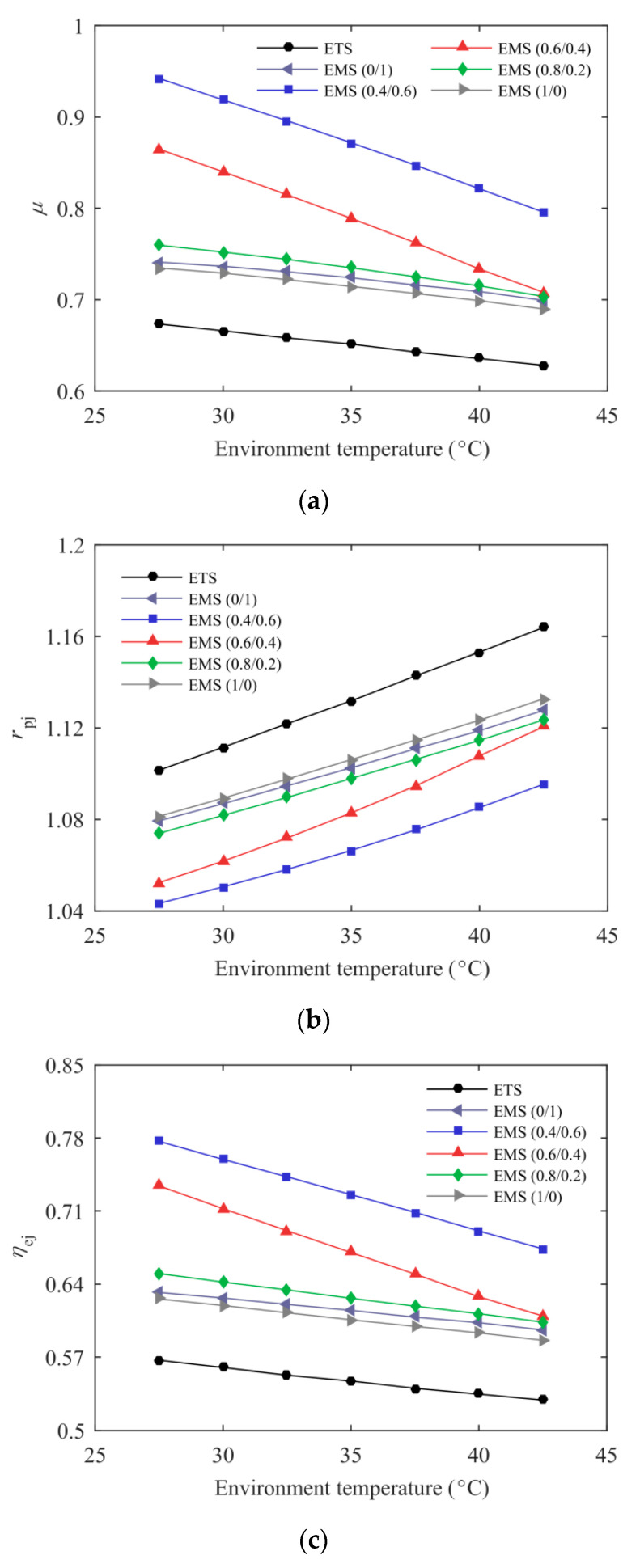
The effect of environmental temperature on the ejector performance: (**a**) entrainment ratio; (**b**) pressure lift ratio; (**c**) ejector exergy efficiency.

**Figure 7 entropy-21-00874-f007:**
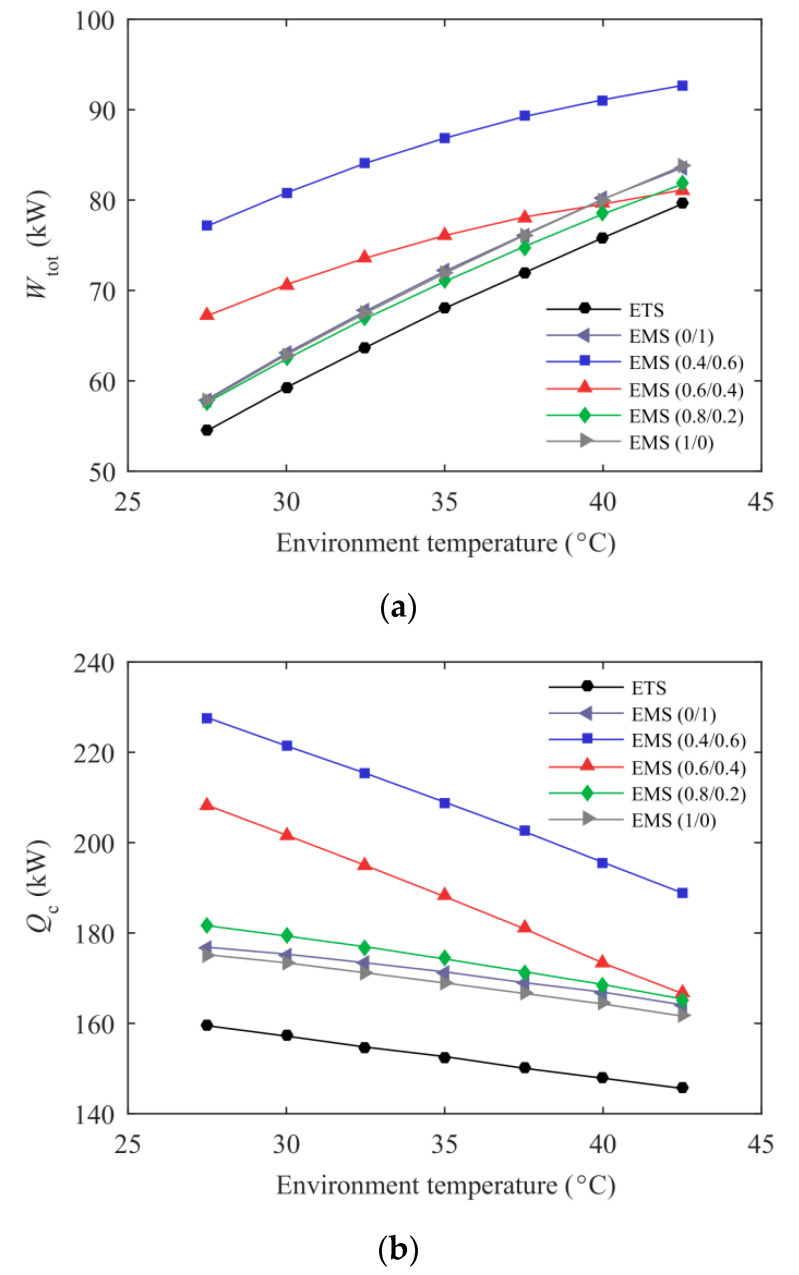
Energetic performance of the ETS and the EMS cycles variation with environmental temperature: (**a**) Total power consumption; (**b**) Cooling capacity; (**c**) Maximum COP_c_.

**Figure 8 entropy-21-00874-f008:**
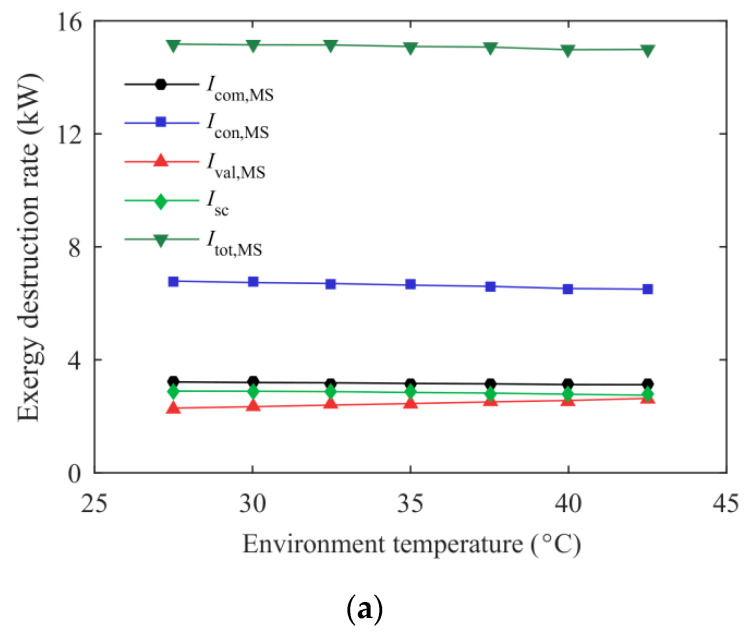
Exergy destruction rate of the MS cycle variation with environmental temperature: (**a**) The EMS (0.4/0.6) cycle; (**b**) The EMS (0.6/0.4) cycle; (**c**) The EMS (0.8/0.2) cycle.

**Figure 9 entropy-21-00874-f009:**
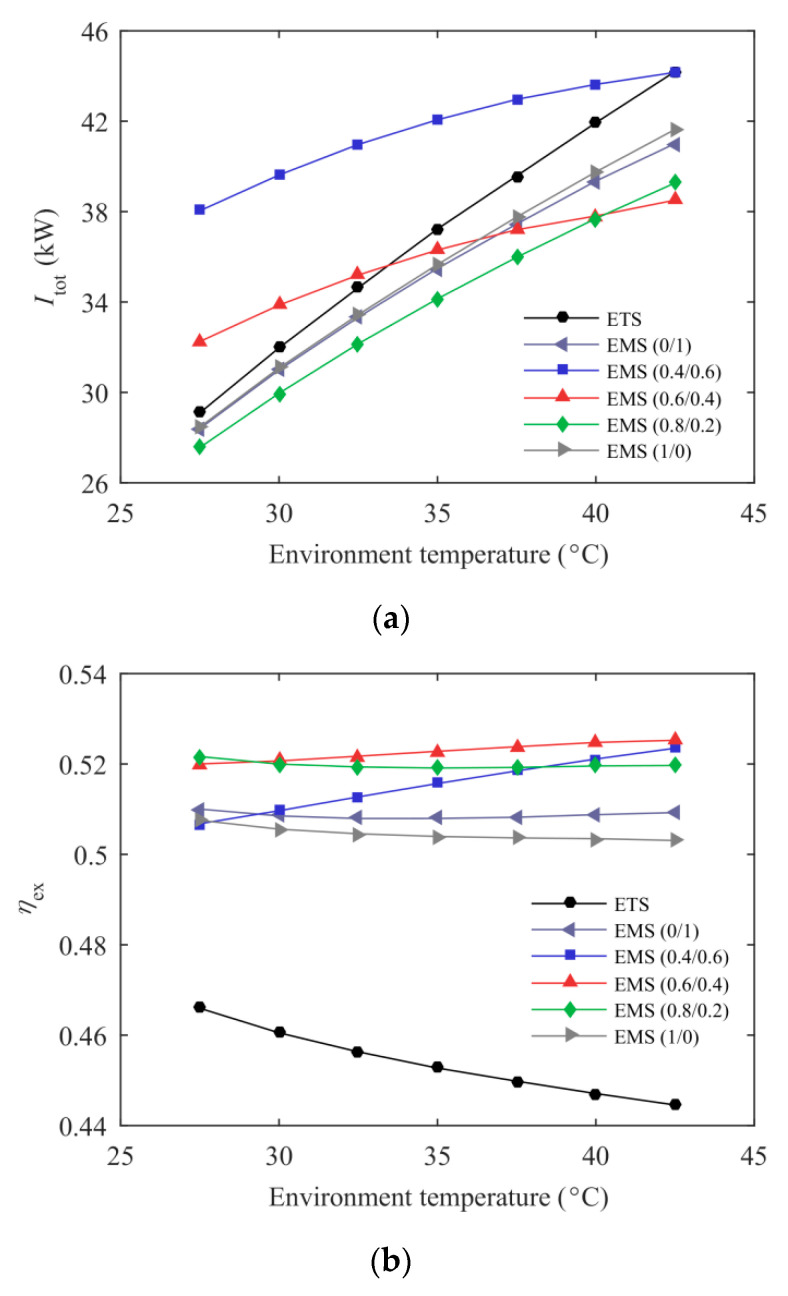
Exergetic performance of the ETS and the EMS cycles’ variation with environmental temperature: (**a**) total exergy destruction rate; (**b**) system exergy efficiency.

**Figure 10 entropy-21-00874-f010:**
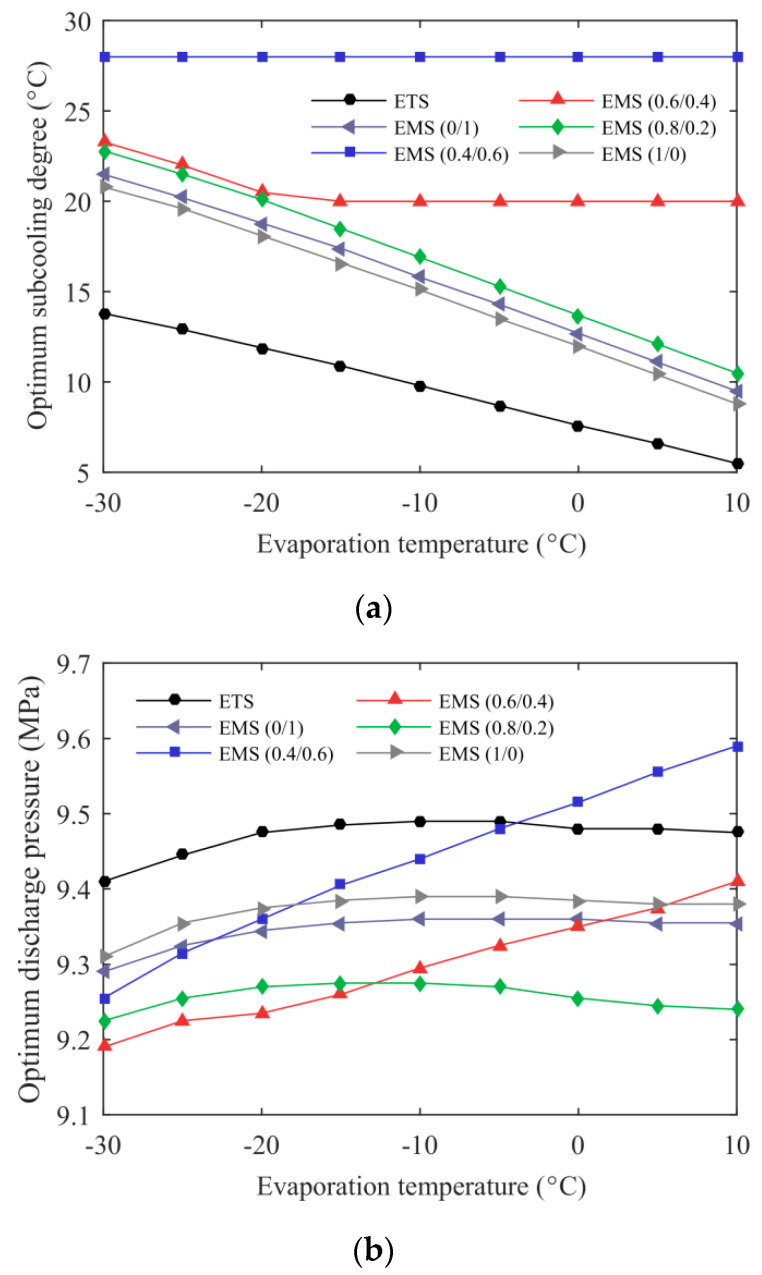
Optimum operating condition of the ETS and the EMS cycles’ variation with evaporation temperature: (**a**) optimum subcooling degree; (**b**) optimum discharge pressure.

**Figure 11 entropy-21-00874-f011:**
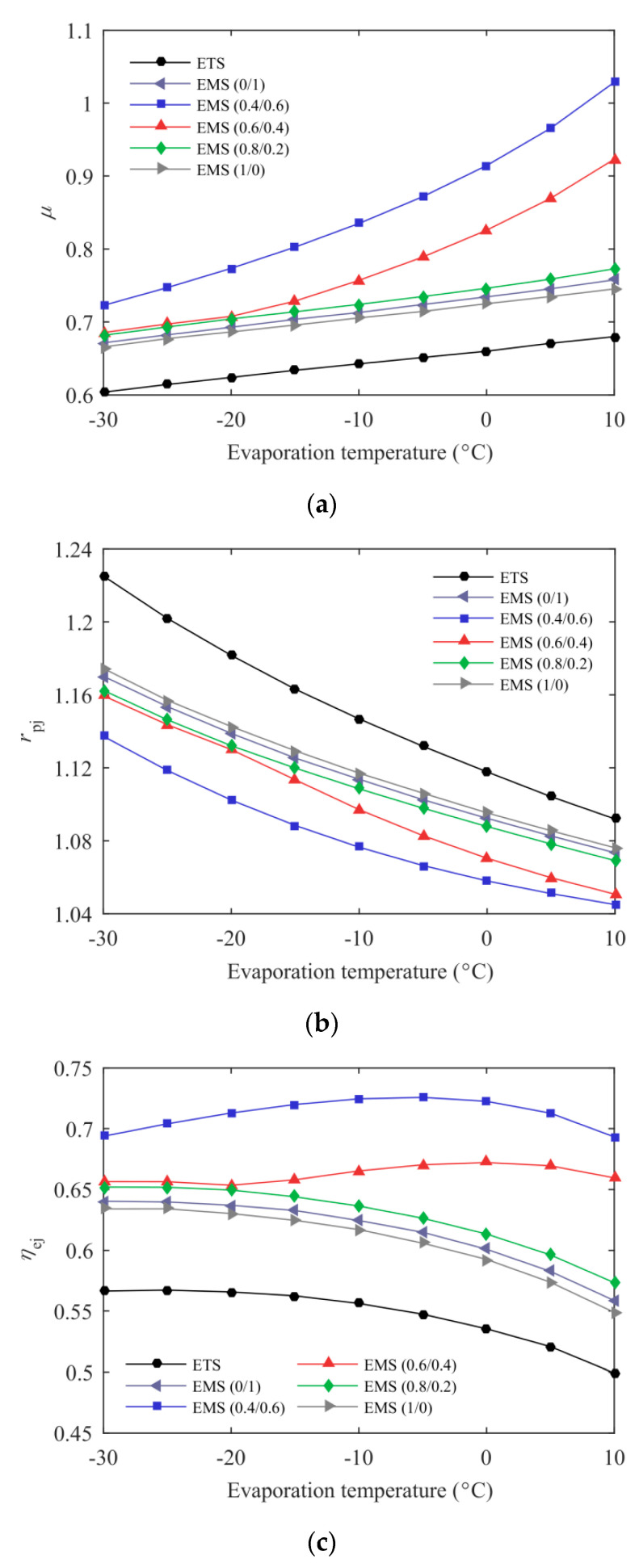
The effect of evaporation temperature on the ejector performance: (**a**) entrainment ratio; (**b**) pressure lift ratio; (**c**) ejector exergy efficiency.

**Figure 12 entropy-21-00874-f012:**
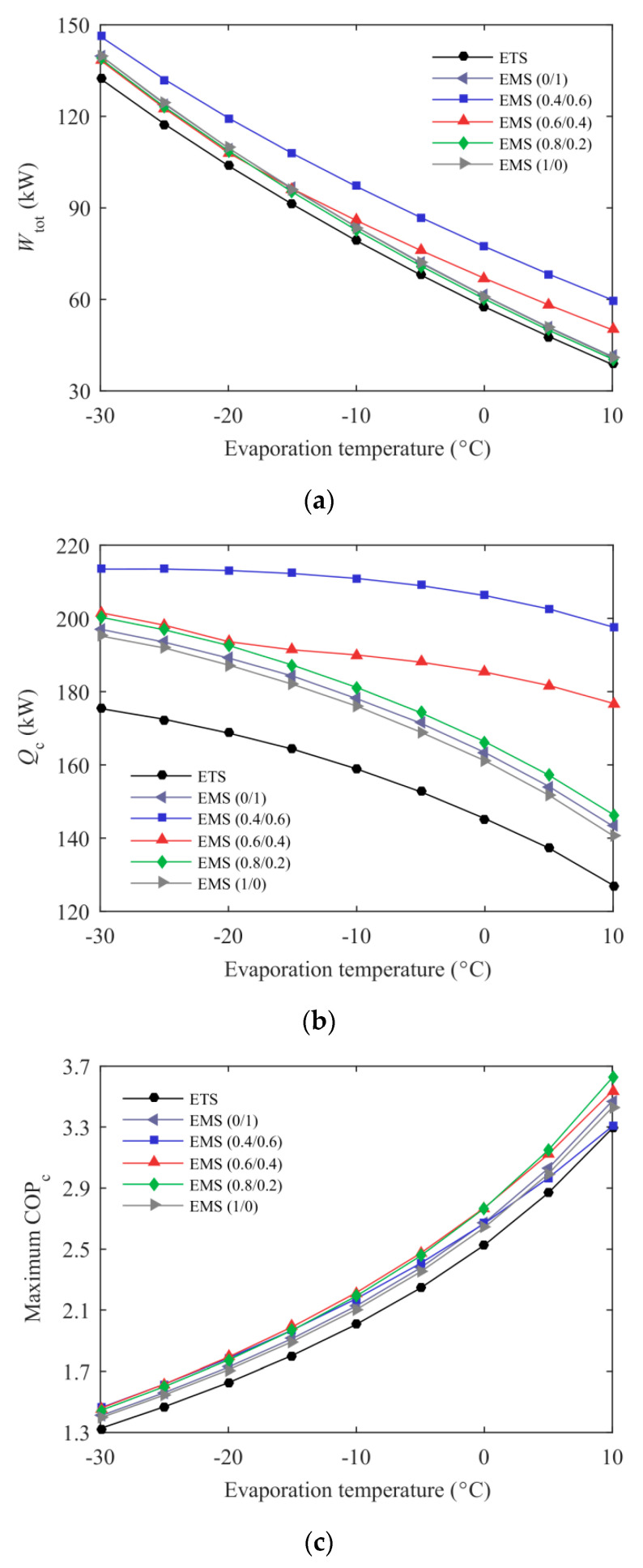
Energetic performance of the ETS and the EMS cycles’ variation with evaporation temperature: (**a**) total power consumption; (**b**) cooling capacity; (**c**) maximum COP_c_.

**Figure 13 entropy-21-00874-f013:**
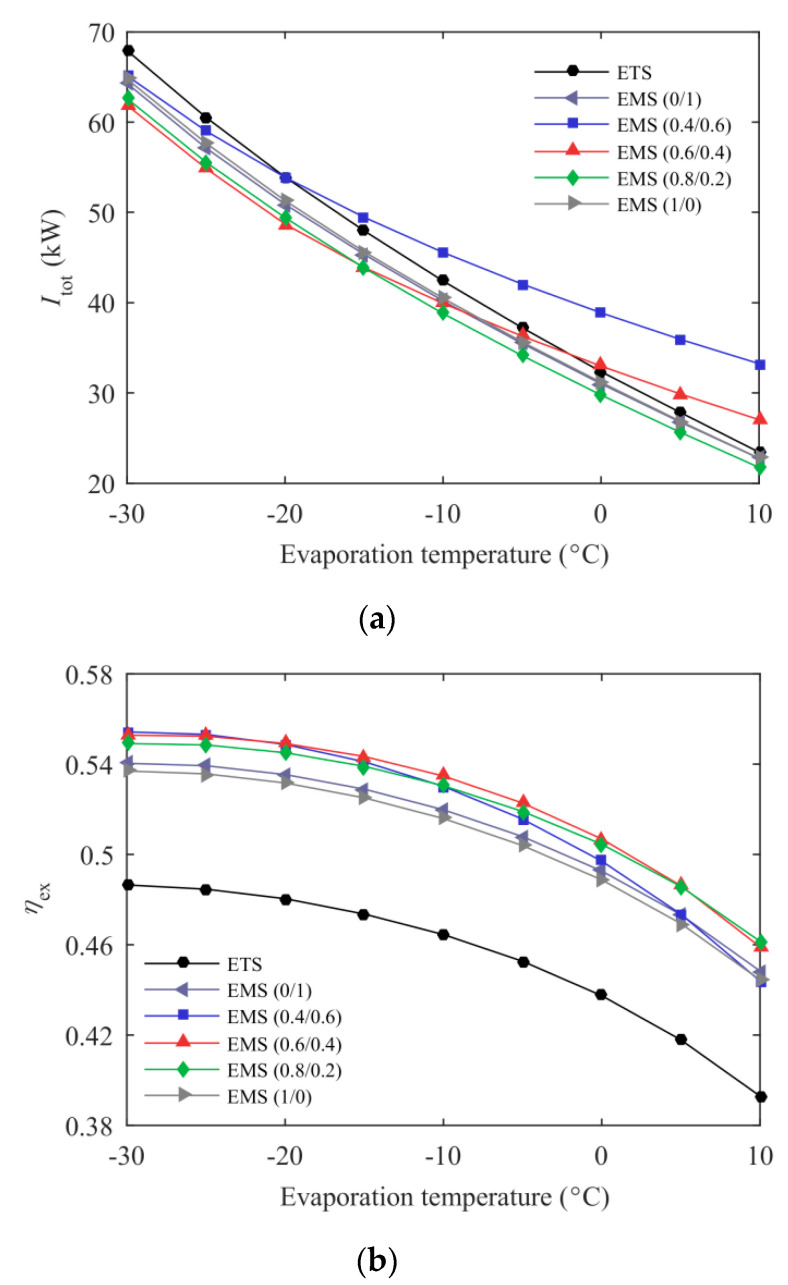
Exergetic performance of the ETS and the EMS cycles’ variation with evaporation temperature. (**a**) Total exergy destruction rate; (**b**) system exergy efficiency.

**Table 1 entropy-21-00874-t001:** The rated operation conditions.

Parameters	Value
Environmental temperature, *T*_0_ (°C)	35
Evaporation temperature, *T*_e_ (°C)	−5
Gas cooler outlet temperature (°C)	*T*_0_ + 5
Temperature of refrigerated object (°C)	*T*_e_ + 5
Air outlet temperature (°C)	*T*_0_ + 8
Pinch temperature difference in the subcooler, Δ*T*_sc,pinch_ (°C)	5
Pinch temperature difference in the condenser, Δ*T*_con,pinch_ (°C)	8
Compressor mechanical efficiency	0.9
Compressor electrical efficiency	0.9
Primary nozzle efficiency	0.8
Mixing efficiency	0.95
Diffuser efficiency	0.8

**Table 2 entropy-21-00874-t002:** Performance comparison at the optimum operating condition.

	ETS	EMS	Increment (%)
*p*_opt_ (MPa)	9.490	9.325	−1.74%
*Q*_c_ (kW)	152.68	188.09	23.19%
Maximum COP_c_	2.24	2.47	10.27%
*η* _ex_	0.45	0.52	15.56%
COP_MS_ (COP_TS_)	5.05	5.73	13.47%
*η* _ej_	0.55	0.67	21.82%

Working condition: *T*_0_ = 35 °C, *T*_e_ = −5 °C, *z* = 0.6.

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
