# Peer review of "Thermodynamic Analysis of Transcritical CO2 Ejector Expansion Refrigeration Cycle with Dedicated Mechanical Subcooling"

_entropy, 2019, doi:10.3390/e21090874_

Round 1

Reviewer 1 Report

From my point of view, I consider the study to show interesting, well-structured and interesting results in the field of refrigeration. I would only recommend mentioning the following work about various configurations in transcritical systems:https://doi.org/10.1016/j.applthermaleng.2016.01.092. In addition, justify the use of equations 19 and 21. The results are linked to these correlations, how would they be affected if other types of regressions or fixed data are considered?

Reviewer 2 Report

  In this paper, transcritical CO2 ejector expansion refrigeration cycle combined with a dedicated mechanical subcooling cycle (EMS) and transcritical CO2 ejector expansion cycle integrated with a thermoelectric subcooling system (ETS) are compared with each other through simulation study. EMS with mixed refrigerants shows the improvement of the coefficient of performance and system exergy efficiency could be up to 10.27% and 15.56%, respectively. However, there are some questions to the authors and corrections to be made.

1.       (page 5 line 150) The authors assumed a couple of things to modelling EMS cycle. However, there is a contradiction. The cycle operates at steady state, and the inlet and outlet velocities in the ejector are neglected. Steady state cycle has constant mass flow rate of refrigerant, and this fact indicates that the inlet and outlet velocities in the ejector are not zero. Therefore, revise the simulation assumption and recalculate.

2.       (page 5 Table 1) Parameters in this table are hard to recognize. The authors use too many subscripts and symbols in schematic diagram. Please revise.

3.       (page 5 Table 1) What is the mixing efficiency? The meaning of it isn’t expressed in this paper.

4.       (page 5 equation 4) The authors simulated the cycle from the viewpoint of energy conservation, and u3 is neglected (Comment 1). However, specific structure of ejector isn’t presented in this paper. Such as, nozzle size, pipe size, etc. To get more precise results, please add the specific structure of ejector.

5.       (page 6 line 171) conversation -> conservation

6.       (page 6 equation 8) I think the left h6 should be changed as h6s

7.       (page 6 equation 10) The authors said, for purpose of ensuring the steady-state operation, the equation 10 must be satisfied. However, the reason was not expressed in this paper. Please add the theoretical background.

8.       (page 6 equation 20) Isentropic efficiency of this cycle is not presented in this paper. Please add it.

9.       (page 10 figure 4) (a) and (b) have different COPc axis and subcooling degree axis. In order for the readers to perceive the difference, unify the axis of them.

10.    (page 11 figure 5) The authors simulates R32/R1234ze(Z) mixed refrigerants at 0.4/0.6, 0.6/0.4, and 0.8/0.2. However, it is difficult to judge the effect of mixed refrigerant with these data. Please add the another cases and single refrigerant case.

11.    (Figure 7 to Figure 12) The description and the figure are separated. Please revise it.

12.    (page 16 Figure 10 (a)) The ETS and EMS (0.6/0.4) data did not show optimum subcooling degree glide on a particular section or whole section. There is a need for further explanation.
